# Identification of an Intravenous Injectable NK1 Receptor Antagonist for Use in Traumatic Brain Injury

**DOI:** 10.3390/ijms25063535

**Published:** 2024-03-21

**Authors:** Robert Vink, Alan Nimmo

**Affiliations:** 1Clinical and Health Sciences, University of South Australia, Adelaide, SA 5001, Australia; robert.vink@unisa.edu.au; 2College of Medicine and Dentistry, James Cook University, Townsville, QLD 4811, Australia

**Keywords:** tachykinin NK1 receptor, traumatic brain injury, NK1 receptor antagonist, blood–brain barrier, cerebral oedema, neuroinflammation

## Abstract

Traumatic brain injuries represent a leading cause of death and disability in the paediatric and adult populations. Moderate-to-severe injuries are associated with blood–brain barrier dysfunction, the development of cerebral oedema, and neuroinflammation. Antagonists of the tachykinin NK1 receptor have been proposed as potential agents for the post-injury treatment of TBI. We report on the identification of EUC-001 as a potential clinical candidate for development as a novel TBI therapy. EUC-001 is a selective NK1 antagonist with a high affinity for the human NK1 receptor (Ki 5.75 × 10^−10^ M). It has sufficient aqueous solubility to enable intravenous administration, whilst still retaining good CNS penetration as evidenced by its ability to inhibit the gerbil foot-tapping response. Using an animal model of TBI, the post-injury administration of EUC-001 was shown to restore BBB function in a dose-dependent manner. EUC-001 was also able to ameliorate cerebral oedema. These effects were associated with a significant reduction in post-TBI mortality. In addition, EUC-001 was able to significantly reduce functional deficits, both motor and cognitive, that normally follow a severe injury. EUC-001 is proposed as an ideal candidate for clinical development for TBI.

## 1. Introduction

Traumatic brain injuries (TBIs) represent the leading cause of death and disability in children, adolescents, and the young adult population [1,2]. However, the mortality rates only represent one part of the picture. The morbidities associated with TBIs place an enormous social and financial burden on society [3]. There is however a significant potential to reduce the impact of TBIs since much of the long-term or permanent damage is not due to the primary injury but rather the delayed, secondary injury mechanisms [4,5]. With diffuse axonal injury, a type of injury commonly associated with motor vehicle and sporting accidents, only a relatively small proportion of axons are damaged at the time of the primary injury. The more significant axonal degeneration occurs over a period of time after the initial insult, providing a window of opportunity for therapeutic intervention [3,5].

Inflammation is a universal response to tissue injury and is known to play a key role in a number of central nervous system (CNS) pathologies, including the response to TBI [6]. An important aspect of the inflammatory response in the brain is blood–brain barrier (BBB) dysfunction [5,7]. Altered BBB function may serve as a precursor for neuroinflammation, which is a significant secondary injury mechanism, and is linked to the development of complications such as cerebral oedema and raised intracranial pressure (ICP) [5,8,9,10]. In relation to developing novel therapies for the management of TBI, altered BBB function has been identified as a potential target [11,12,13].

Studies in peripheral tissues and organs provide a wealth of data to support the notion that perivascular nerves, and the neuropeptides they release, may play an important role in mediating vascular inflammatory responses, including oedema [14,15,16,17,18]. As a result, there was an interest in whether similar mechanisms may be involved in regulating BBB integrity and CNS inflammation [19,20,21]. An initial study, involving the use of capsaicin as a pre-injury treatment to produce neuropeptide depletion, was able to demonstrate that in a rodent model of TBI, there was a significant reduction in BBB dysfunction and oedema formation following injury, together with a significant decrease in the motor and cognitive deficits normally associated with severe injury [19]. From this initial observation, further studies using selective receptor antagonists were performed to assess whether the post-injury administration of an antagonist could exert a similar protective effect.

From knowledge of peripheral inflammatory mechanisms, the pro-inflammatory tachykinin peptide, substance P (SP) was identified as a potential target for post-injury treatment [16,18]. Clinical data also support a role for SP in TBI, with strong correlations between SP levels and patient outcomes [22,23,24]. There are three primary subtypes of the tachykinin receptors, namely NK1, NK2 and NK3 receptors. Whilst there is a potential for cross-reactivity with the endogenous ligands and the different receptor subtypes, SP binds preferentially to the NK1 receptor (NK1R), whilst neurokinin A and neurokinin B bind preferentially to the NK2 and NK3 subtypes, respectively [25]. As a result, the NK1R was considered an appropriate target to antagonize the actions of SP. Studies using the post-injury administration of an NK1R antagonist in multiple animal models of TBI indicated that post-injury treatment with an antagonist could, in line with the effect of pre-injury capsaicin treatment, ameliorate the acute inflammatory response, and significantly improve functional outcome [26,27,28,29,30]. Importantly, the window for therapeutic intervention was consistent with clinical response times [27,31]. For a number of these studies, N-acetyl-L-tryptophan (NAT) was used as an agent with assumed NK1 receptor antagonist activity, primarily based on the reported activities of analogues of L-tryptophan [26,32]. However, questions have been raised about the use of NAT, with a recent publication demonstrating a lack of NK1R binding by NAT [33]. In contrast, an in silico screening assay for NK1R antagonist activity predicted that NAT would have favourable binding characteristics [34]. Whilst a question mark remains around the specific mechanism of action of NAT, the protective effects seen in these studies results are replicated when a highly selective NK1 antagonist is administered [30,35,36,37]. These successful preclinical studies led to the process of identifying a potential clinical candidate for use in TBI. Given both the severe and acute nature of TBI, it was considered that for an NK1 antagonist to be clinically effective, it would need to be available for administration in an intravenous (*iv*) formulation. However, one characteristic that is shared by many NK1R antagonists is that they have relatively low aqueous solubility, a characteristic that is influenced by the lipophilic nature of the SP-binding site on the NK1R [38,39,40].

Due to their strong anti-emetic actions, NK1R antagonists have become an important tool in the management of chemotherapy-induced nausea and vomiting (CINV) [41,42]. Although originally developed for the treatment of anxiety and depression, aprepitant (Emend; MK-869; Merck, Rahway, NJ, USA) was the first NK1R antagonist to gain approval for use in CINV [41,43]. Subsequently, two other agents, rolapitant and netupitant, have also been approved for the same indication [44,45]. All three agents were initially only available in oral formulation. However, in terms of managing CINV, it can be advantageous to be able to administer an anti-emetic agent intravenously [46]. As a parent drug, aprepitant has low aqueous solubility (0.2 µg/mL) and hence is not suitable for iv formulation [46]. One approach to overcome low aqueous solubility is through the development of a prodrug [47]. Following the introduction of the oral form of aprepitant, an iv-injectable prodrug, fosaprepitant (Emend for injection; MK-0517), was subsequently developed [48,49]. Fosaprepitant is a water-soluble, phosphoryl prodrug of aprepitant, which, following administration, is converted to aprepitant through the action of phosphatases [49]. Fosaprepitant is administered by slow infusion (20–30 min) in order to avoid venous irritation and is converted to aprepitant within 30 min of the end of infusion [46,49]. The development of a phosphoryl prodrug has been subsequently utilized for other NK1 antagonists, most notably fosnetupitant [50]. In contrast, an iv formulation of rolapitant was developed that involved generating an emulsion using polyoxyl 15 hydroxystearate [51]. Unfortunately, a significant risk of anaphylaxis and severe hypersensitivity reactions was observed with this formulation, leading to its production and use being suspended [51,52].

Phosphoryl prodrugs, such as fosaprepitant, represent effective, iv-infusible agents. However, the requirement for infusion and subsequent metabolic alteration induces a delay in achieving peak plasma levels, which negates one of the common benefits of intravenous administration, namely a rapid onset of action. In relation to managing a severe TBI, the need for prodrug metabolism may also introduce another variable. It is known that severe trauma can result in significant metabolic changes in a patient [53]. The liver plays an important role in the inflammatory response to TBI, and with severe injury, this may be excessive [54]. The associated acute-phase reaction, and release of acute-phase proteins, can lead to leukocyte infiltration in organs distant from the injury site, including the liver itself [55]. This may lead to hepatocellular injury as well as contribute to multiple organ dysfunction [54,55]. As a result, it is possible that prodrug pharmacokinetics may change in the trauma situation.

Based on these issues, it was considered that the optimal agent for clinical use in TBI would be an NK1 antagonist that was capable of aqueous iv formulation as the active parent drug, whilst also having good brain penetration. Hoffmann-La Roche (Basel, Switzerland), like a number of other pharmaceutical companies, had an interest in developing NK1R antagonists for their potential use in depression and emesis. The screening program undertaken led to the development of a number of selective, orally active NK1 antagonists, including netupitant [56]. Importantly, given the species differences in NK1 receptors, these agents were screened for activity against the human NK1R [57]. Given the range of analogues developed by this screening program, it was decided to perform a retrospective data mine into this bank of compounds to see whether any of the agents possessed the pharmacodynamic and physicochemical properties that would make them suitable for development as a clinical agent for TBI. Interrogation of the available data revealed that one compound (RO-0671721; EUC-001) possessed the required characteristics. The present study describes the pre-clinical characterization of EUC-001 as a potential candidate for development as a treatment for TBI.

## 2. Results

Data interrogation of the bank of NK1R antagonists developed as part of the Roche screening program [56] indicated that one compound, *N*-(3,5-Bis-trifluoromethyl-benzyl)-*N*-methyl-6-(4-methyl-piperazine-1-yl)-4-o-tolyl-nicotinamide (EUC-001), possessed the appropriate physicochemical characteristics required (Figure 1). EUC-001 was found to have an aqueous solubility of 0.5 mg/mL as a free base, whilst the hydrochloride salt could be dissolved at a concentration of 1.5 mg/mL, indicating the potential to administer it in an iv formulation. Whilst EUC-001 has greater aqueous solubility than many NK1 antagonists, it was found to still possess good CNS penetration, with a brain/plasma ratio of 3/7.

EUC-001 was characterized both in vitro and in vivo for its affinity and selectivity in terms of competitive antagonism of the NK1R, as well as for its potential efficacy in an animal model of TBI.

### 2.1. In Vitro Characterization of EUC-001

The affinity of EUC-001 for the human NK1R was assessed by radioligand binding assays using Chinese hamster ovary (CHO) cells transfected with the human NK1R. Radioligand binding studies, evaluating competition for [^3^H] SP binding (0.6 nM), indicated that EUC-001 had a high affinity for the recombinant human NK1R, with a Ki of 5.75 × 10^−10^ M, whilst its affinity for recombinant human NK2 and NK3 receptors was significantly less, with a Ki of 3.4 × 10^−6^ M and 1.7 × 10^−5^ M, respectively.

In vitro functional assays were performed by assessing changes in free Ca^2+^ ion concentration in human glioblastoma (U373MG) cells, which endogenously express the NK1R. Agonist activity of EUC-001 is expressed as a percentage of the agonist response induced by the selective NK1R agonist [Sar^9^, Met(O_2_)^11^]-substance P (100 nM; Figure 2A), whilst antagonist activity is expressed as a percentage change in the response induced by 3 nM [Sar^9^, Met(O_2_)^11^]-SP (Figure 2B). EUC-001 did not exhibit any detectable agonist activity (Figure 2A), whilst it was found to be a potent inhibitor of the [Sar^9^, Met(O_2_)^11^]-SP-mediated response, with an IC_50_ of 6.9 × 10^−10^ M (Figure 2B).

The pharmacological specificity of EUC-001 was assessed using the commercially available selectivity profiling services provided by Cerep (Celle l’Evescault, France). Initially, the activity of EUC-001, at a fixed concentration of 10 μM, was evaluated in a range of assays to assess pharmacological selectivity relative to other cell-surface receptors, ion channels and transport proteins. Where any significant binding was detected, further assays were performed to quantify potential non-selective activity. EUC-001 exhibited low-affinity antagonist activity for some human G-protein-coupled receptors (Table 1). In terms of this non-specific activity, the IC_50_s are approximately 3.6 log units greater than that required for NK1R antagonism.

### 2.2. In Vivo Characterization of EUC-001

The ability of EUC-001 to antagonize central NK1Rs in vivo was examined using the gerbil foot-tapping response. The hind-foot-tapping response is a species-specific, CNS-mediated reaction that occurs in response to fear-evoking situations or aversive stimuli [58]. The same, stereotyped response can be induced by the intracerebroventricular (*icv*) injection of a selective NK1 receptor agonist, such as GR73632 [59]. In turn, this NK1R-mediated response can be inhibited by centrally acting NK1R antagonists, as well as other anxiolytic agents [59]. Animals were pretreated with a range of concentrations of EUC-001 2 h prior to the icv administration of the NK1R agonist, GR73632. Administration of EUC-001 resulted in a dose-dependent blockade of the foot-tapping response (Figure 3). For oral administration (*po*), the IC_50_ dose for the inhibition of this response was 0.89 mg/kg, whilst iv administration gave an IC_50_ of 0.68 mg/kg.

### 2.3. Effect of EUC-001 in a Rodent Model of TBI

To assess whether a highly selective NK1R antagonist could replicate the protective effects previously observed with NAT, the efficacy of EUC-001 was assessed in a rodent model of TBI [26,31]. Injury was induced using an impact-acceleration model of TBI which generates a rodent form of diffuse axonal injury, a major feature of clinical head injury [29,60]. The level of injury induced in these studies was classified as “severe” [60]. In terms of assessing treatment effects, the study was performed in a blinded manner. In addition to EUC-001, the glutamate antagonist, MK801, being a putative neuroprotective agent, was included as a positive control [61]. In line with the “severe” nature of the injury, significant mortality rates were observed within the cohort of injured animals. However, the mortality rates varied between the treatment groups. In the “drug vehicle” group, a mortality rate of 30% was observed, which is in line with the nature of the injury. Treatment with MK801 (10 mg/kg) 30 min after injury resulted in a 48% mortality rate, which, although an unexpected increase, was not significantly different from the drug vehicle group. In contrast, no mortality was observed in the group treated 30 min post-injury with EUC-001 (10 mg/kg).

#### 2.3.1. Effect of EUC-001 on BBB Permeability following TBI

One of the major impacts seen with traumatic brain injury is impaired BBB function. To assess the potential effects of EUC-001 on BBB function, animals were subject to a severe TBI, which resulted in a marked increase in BBB permeability as evidenced by Evan’s blue extravasation in animals treated with drug vehicle alone (Figure 4A). Treatment 30 min post-injury with EUC-001 resulted in a significant reduction in Evan’s blue leakage (*p* < 0.001) relative to drug vehicle, whilst treatment with MK801 resulted in a small decrease in Evan’s blue extravasation (<0.05). Following this result, the effect of administering a range of concentrations of EUC-001 by bolus iv injection 30 min after injury was assessed by quantifying changes in Evan’s blue extravasation within the brain. EUC-001 was found to produce a dose-dependent decrease in Evan’s blue extravasation (Figure 4B), with an IC_50_ of 0.43 mg/kg.

The BBB dysfunction that follows acute CNS injuries can contribute to significant complications, such as cerebral oedema and raised intracranial pressure (ICP) [21]. The development of cerebral oedema was assessed using the wet weight–dry weight method, with tissues collected 6 h post-injury. Following TBI, treatment with drug vehicle resulted in a significant (*p* < 0.001) increase in brain tissue water content, whilst in the group treated 30 min post-injury with EUC-001, the tissue water content was not significantly different from sham (prepared but uninjured) animals (Figure 5).

#### 2.3.2. Effect of Treatment with EUC-001 on Functional Outcomes following TBI

For those who survive the acute phase of a severe TBI, the injuries commonly result in long-term or permanent neurological dysfunction, affecting either, or both, cognitive and motor function [3]. To assess whether post-injury treatment with EUC-001 could also help ameliorate functional deficits, the effect of treatment on functional outcomes was assessed. For this study, the effect of injury, and post-injury treatment, on motor and cognitive function was assessed using the rotarod test and Barnes maze, respectively [62,63,64]. All animals were pretrained on both functional outcome tests twice per day over 5 days before injury to establish a normal, uninjured baseline (Day 0).

The rotarod test is considered the most sensitive measure of motor outcome after rodent TBI [64]. The rotational speed of the device was increased from 0 to 30 r.p.m., and the duration (in seconds) where the animals either successfully completed the task or failed to walk actively was recorded as the rotarod score. The maximum duration of the task was 120 s. Animals treated with drug vehicle alone exhibit a rapid post-injury decline in motor function, with only a partial recovery of this function over the subsequent days (Figure 6). A similar picture was seen with animals treated 30 min post-injury with MK801 (10 mg/kg). However, animals treated post-injury with EUC-001 (10 mg/kg) exhibited a much smaller decline in motor function in the immediate post-injury phase whilst exhibiting only minor deficits towards the end of the study period (Figure 6).

Cognitive outcome was assessed using the Barnes maze in order to reduce the potential confounding effects that may be associated with tests such as the Morris water maze [62]. In the Barnes maze, animals were placed under a cover in the centre of an elevated 1.2 m diameter board containing 19 holes around the periphery. One hole contained the entrance to a darkened escape tunnel that was not visible from the surface, relying on the animal to use surrounding visual cues to locate the tunnel. The time, in seconds, for the animal to locate and enter the escape tunnel was recorded. Following pre-training, the effect of injury, and post-injury treatment, on the time taken for the animal to locate the tunnel was measured (Figure 7). For animals treated post-injury with EUC-001 (10 mg/kg), there appeared to be minimal impact on the performance time associated with this task. In contrast, animals treated with drug vehicle alone exhibited delays in locating the escape tunnel. In this test, post-injury treatment (30 min) appeared to provide some improvement in performance over the duration of the study. The effect of post-injury treatment with EUC-001 was found to be statistically significant on half of the test days when compared to drug vehicle (Figure 7).

## 3. Discussion

When trying to deal with clinical trauma situations, such as TBI, there is a tendency to assume that most of the problem will result from the primary injury. However, it has long been recognized that the delayed secondary injury processes play a key role in determining patient outcomes, both in terms of mortality and morbidity [5]. As such, there has been a significant interest in developing interventions that may help ameliorate these secondary injury responses [65]. Inflammation is a universal response to tissue injury and is considered to play a key role in TBI, contributing to altered BBB function and associated complications, such as cerebral oedema and raised intracranial pressure [6,7,66]. Indeed, there appears to be a two-way relationship between BBB dysfunction and neuroinflammation. Altered BBB function may serve as a precursor for neuroinflammation, while neuroinflammation will impact BBB permeability and function [67]. Hence, in terms of identifying a validated target for the development of a novel TBI therapy, altered BBB function, and the role inflammation plays in this, is well supported by both scientific and clinical evidence [11,12].

Research on mechanisms of peripheral inflammation has provided considerable evidence to support a role for SP and the NK1R [68,69]. In addition to mediating vascular responses, they appear to play a broader role in the innate immune response [70]. SP increases vascular permeability and promotes oedema [71]. SP also enhances leukocyte migration through its effects on vascular and intra-cellular adhesion molecules, as well as matrix metalloproteinase (MMP) secretion [16,72]. SP is one of the earliest activators of nuclear factor-κB (NFκB) in response to injury and can increase the secretion of other pro-inflammatory mediators, including interleukin (IL)-1β, IL-6, and tumor necrosis factor-α (TNF-α), elaborating the inflammatory response [70]. Hence, rather than just being a mediator associated with oedema formation, the evidence suggests that SP may play a broader role in inflammation [8]. For example, studies suggest a role for SP in neutrophil migration and activation at the site of inflammation [73].

Over the past 40 years, the NK1R has been a focus of much research in relation to its potential value as a clinical target. Given the anatomical localization of SP, much of that initial focus was on whether NK1 antagonists could exert analgesic actions [74]. Whilst they do appear to exert effects in relation to the response to noxious stimuli, it appears that their action is more at the level of altering behavioural responses to stressful stimuli, rather than producing clinical analgesia [74,75]. However, this research led to the discovery that NK1 antagonists have powerful anti-emetic actions. As a result, NK1 antagonists are now established as safe and effective clinical agents, primarily through their use in the treatment of CINV [76,77].

The natural ligand for the NK1R is the neuropeptide, SP. However, in terms of optimizing a clinical agent, the drug development and manufacturing process favours non-peptide small molecules [78]. Multiple pharmaceutical companies were active in trying to develop non-peptide ligands, with Snider and colleagues being the first to report on the successful development of a selective, non-peptide NK1 antagonist, CP-96,345 [79]. This success was followed by that of other companies [32,56]. However, the development of safe and effective clinical agents was not without its challenges. There are significant species differences in the NK1R, and these impact upon antagonist, rather than agonist, activity; hence, potential agents need to be screened for activity against the human NK1R [57]. Some NK1R antagonists, such as CP-96,345, were found to interact with Ca^2+^ ion channels, although this problem was circumvented with the subsequent development of CP-99,994 [80]. In terms of clinical development, the introduction of aprepitant was a key milestone, although this was not without its challenges, both in relation to the synthesis pathway and achieving good bioavailability of the oral product [81]. Hence, all these factors have to be taken into consideration in the development of a novel clinical candidate.

In terms of its development, EUC-001 was developed against the recombinant human NK1R expressed in CHO cells. EUC-001 exhibits a high affinity for the human receptor, with a Ki of 5.75 × 10^−10^ M. In vitro functional assays involving human glioblastoma (U373MG) cells, which endogenously express the NK1R, indicated that EUC-001 had no agonist activity, but that it was a potent antagonist of the receptor, inhibiting the [Sar^9^, Met(O_2_)^11^]-SP-induced response, with an IC_50_ of 6.9 × 10^−10^ M. EUC-001 also exhibited potent antagonist activity in vivo, as evidenced by its ability to inhibit the GR73632-induced foot-tapping response in gerbils. The NK1 receptor pharmacology in gerbils is considered to be representative of humans [59]. EUC-001 was able to produce a dose-dependent blockade of the foot-tapping response when it was administered either orally (IC_50_ = 0.89 mg/kg) or intravenously (IC_50_ = 0.68 mg/kg). Hence, the drug is not only active in vivo, but it exhibits its activity within the CNS.

Importantly, EUC-001 also exhibits a high degree of selectivity when screened for activity against a range of cell-surface receptors, transport proteins and ion channels, including Ca^2+^ ion channels. When any significant binding was detected during initial screening assays, further assays were performed to quantify potential non-selective agonist or antagonist activity. EUC-001 exhibited low-affinity antagonist activity for some human G-protein-coupled receptors, including some subtypes of serotonergic, dopaminergic, and cholinergic receptors (Table 1). However, analysis of this non-specific activity revealed that the IC_50_ for antagonism of these receptors were between 3.6 and 4.6 log units greater than that required for NK1R antagonism. Hence, any non-selective binding to these receptors would be negligible with the EUC-001 concentrations required for NK1R antagonism.

In contrast to many NK1 antagonists, EUC-001, particularly in the form of the hydrochloride salt, has reasonable aqueous solubility (1.5 mg/mL), making it a potential candidate for iv formulation. For all the in vivo experiments in this study, the drug was prepared in the form of an isotonic aqueous solution and administered as a slow bolus injection. Efficacy studies were performed in a blinded manner, with the glutamate NMDA antagonist, MK801, being included as a positive control. MK801 has been shown to exert neuroprotective effects in animal models of TBI, although the time window for efficacy is small [61].

Treatment with EUC-001 30 min post-injury was able to significantly reduce BBB permeability, as evidenced by Evan’s blue leakage into the brain, as compared to drug vehicle alone or post-injury treatment with MK801 [19,26,27,28]. This action of an NK1 antagonist on the cerebral vasculature is in line with what has been widely observed in peripheral tissues [68,82,83]. That ability of EUC-001 to inhibit Evan’s blue leakage following TBI occurred in a dose-dependent manner with the IC_50_ being 0.43 mg/kg, which correlates closely with the IC_50_ for the inhibition of the gerbil foot-tapping response when the drug is administered intravenously (0.68 mg/kg). EUC-001 has a brain/plasma distribution ratio of 3/7, and that inhibition of the foot-tapping response certainly reflects CNS activity. In contrast, the BBB sits at the interface of the CNS and peripheral circulation and hence may be influenced by both circulating and brain concentrations [84].

EUC-001 was also able to significantly ameliorate the development of cerebral oedema, as determined by the wet weight–dry weight method following TBI. Cerebral oedema is considered to be a leading cause of in-hospital mortality with TBI, and the capacity for EUC-001 to ameliorate cerebral oedema could correlate with the marked reduction in mortality observed in this study (0% EUC-001 vs. 30% drug vehicle). Currently, decompressive craniectomy (DC) is an accepted approach to dealing with high ICP that is refractory to medical management [9]. Whilst DC may reduce mortality in these situations, there are still concerns about long-term morbidities [9]. In a sheep model of stroke, the administration of an NK1R antagonist has been shown to be as effective in managing raised ICP as the much more invasive approach of DC [36].

However, as has been observed with DC, simply managing oedema and reducing ICP is not, in itself, a way of improving the long-term outcomes for patients following a severe TBI [9]. In addition to managing the acute issues, there is also a need to address the problem of the long-term functional deficits that can accompany a TBI. The effect of post-injury administration of EUC-001 was assessed in relation to both motor and cognitive functional outcomes. EUC-001 was found to significantly preserve the level of motor function following a severe TBI as compared to both drug vehicle and MK801, with animals exhibiting close-to-normal motor function within 5 days of the injury. The post-injury administration of MK801 did not provide any beneficial outcome relative to the drug vehicle. Whilst NMDA antagonists, like MK801, have been shown to exert neuroprotective effects, the beneficial effects are primarily seen when they are administered immediately before or after injury [61,85]. Glutamate is a key excitatory transmitter in the CNS. However, events like trauma or cerebral ischaemia can lead to excessive glutamate release, leading to an excitotoxic action, which can be ameliorated by an NMDA antagonist [86]. However, glutamate appears to fulfil a dual role after injury and may play a role in the CNS recovery process [87]. This has been postulated as one of the reasons that clinical trials of NMDA antagonists in TBI and stroke have failed [87,88]. In contrast, an SP antagonist has been shown to exert its beneficial effects on neurological outcomes when administered up to 12 h post-injury [31]. For both drug vehicle and MK801 groups, there was a partial (approximately 50%) recovery of motor function over the 10-day test period, although this may be a function of the testing paradigm. In a manner akin to post-injury rehabilitation, the daily assessment of motor function can promote more functional recovery as opposed to weekly assessments performed over a longer timeframe [64].

In addition to helping to preserve motor function, EUC-001 also exerted positive effects in relation to cognitive function. Whilst the differences between treatment groups were not as marked as those seen in relation to motor function, the performance of EUC-001-treated animals on the Barnes maze was close to pre-injury levels for the duration of the study. Statistical significance relative to drug vehicle-treated animals was observed on five of the test days. Unlike its effects on motor function, MK801 did appear to have a beneficial action on cognitive function, with performance returning close to pre-injury by the end of the study period. This beneficial effect on cognitive function, without a parallel effect on motor function has also been observed with amantadine [89]. Amongst its other actions, amantadine is an inhibitor of NMDA receptors [90].

The results of the current study indicate that the post-injury administration of a selective, high-affinity NK1R antagonist can help maintain BBB integrity and ameliorate post-injury cerebral oedema following a severe TBI. In addition, EUC-001 was also found to confer longer-term functional benefits, both in relation to motor and cognitive function.

## 4. Materials and Methods

All experimental protocols were approved by the Experimental Ethics Committees of James Cook University (A490), the University of Adelaide (M41-2003) and the Institute of Medical and Veterinary Science (20/03). All studies were conducted according to guidelines established for the use of animals in experimental research as outlined by the Australian National Health and Medical Research Council.

### 4.1. In Vitro Radioligand Binding Assays

Radioligand binding assays were performed using CHO cells (ATCC, Middlesex, UK) transfected with the human NK1R (accession number NM_001058). Assays were performed in 50 mM HEPES buffer (pH 7.4; Sigma-Aldrich, St Louis, MO, USA) containing BSA (0.04%; Sigma-Aldrich, St Louis, MO, USA), leupeptin (8 μg/mL; Sigma-Aldrich, St Louis, MO, USA), MnCl_2_ (3 mM; Sigma-Aldrich, St Louis, MO, USA) and phosphoramidon (2 μM; Sigma-Aldrich, St Louis, MO, USA). In total, 250 μL of membrane suspension was incubated with 125 μL of unlabelled EUC-001 (synthesized by Hoffmann-La Roche, Basel, Switzerland) dissolved in assay buffer, and 125 μL of [^3^H]SP (final concentration 0.6 nM; Amersham Biosciences, Amersham, UK). Displacement curves were determined with seven concentrations of EUC-001. Following incubation (60 min at room temperature), the mixture was rapidly filtered through pre-soaked Whatman GF/C filters (Sigma-Aldrich, St Louis, MO, USA) and washed with 2 × 2 mL HEPES buffer (Sigma-Aldrich, St Louis, MO, USA), and bound radioactivity was measured by scintillation counting (Beckman Coulter, Brea, CA, USA). Assays were performed in triplicate in 2 separate experiments.

### 4.2. Pharmacological Selectivity Studies

The pharmacological specificity of EUC-001 was assessed using the commercially available selectivity profiling services provided by Cerep (Celle l’Evescault, France). For initial screens, the binding of EUC-001 (synthesized by Hoffmann-La Roche, Basel, Switzerland) at a fixed concentration of 10 μM was evaluated in a range of assays to assess pharmacological selectivity relative to other cell-surface receptors, ion channels and transport proteins.

### 4.3. In Vitro Functional Assay

Human glioblastoma astrocytoma (U373MG) cells (ATCC) were used for in vitro function studies involving intracellular calcium measurements. Cells were routinely grown as monolayers in DMEM/Ham’s F12 (1:l) medium (Gibco, Waltham, MA, USA ) supplemented with 10% fetal calf serum (Gibco, Waltham, MA, USA), 2 mM glutamine, penicillin, and streptomycin (60 pg/mL each) in a humidified CO_2_ atmosphere (10%) at 37 °C. Intracellular calcium concentrations were determined by using the calcium chelating agent fura-2 (Molecular Probes, Waltham, MA, USA). Aliquots of fura-2-loaded cells were subject to agonist stimulation and fluorescence emission was recorded using an MS-5 spectrophotometer (Perkin Elmer, Shelton, CT, USA). Loading of cells, fluorescence monitoring, calibration procedures, and other experimental details were as previously described [91,92].

For agonist activity, results are expressed as a percentage of the control agonist response induced by the selective NK1R agonist [Sar^9^, Met(O_2_)^11^]-substance P (100 nM) for a range of concentrations of EUC-001. Antagonist activity is expressed as a percentage change in the response induced by 3 nM [Sar^9^, Met(O_2_)^11^]-SP (R&D systems) across a range of concentrations of EUC-001. The EC_50_ values (concentration producing a half-maximal specific response) and IC_50_ values (concentration causing a half-maximal inhibition of the control specific agonist response) were determined by non-linear regression analysis of the concentration–response curves generated with mean replicate values using Hill equation curve fitting (Y = D + [(A − D)/(1 + (C/C50)nH)], where Y = specific response, D = minimum specific response, A = maximum specific response, C = compound concentration, and C_50_ = EC_50_ or IC_50_, and nH = slope factor). This analysis was performed using custom software developed at Cerep and validated by comparison with data generated by the commercial software SigmaPlot^®^ 4.0 for Windows^®^ (SPSS Inc., Chicago, IL, USA). For the antagonists, the apparent dissociation constants (K_B_) were calculated using the modified Cheng Prusoff equation (K_B_ = IC_50_/(1 + (A/EC_50_A)), where A = concentration of reference agonist in the assay, and EC_50_A = EC_50_ value of the reference agonist).

### 4.4. Gerbil Foot-Tapping Respon

Female Mongolian Gerbils weighing 30–60 g were administered drug vehicle or a predetermined dose of EUC-001 either orally (*po*) at a volume of 5 mL/kg or intravenously (iv) at a volume of 1 mL/kg. Four hours after *po* administration or immediately after iv administration, each gerbil was anesthetized by inhalation of an oxygen–isofluorane mixture (Ohmeda, West Yorkshire, UK) using a VMC anesthesia machine (Matrix Medical, Orchard Park, NY, USA), after which the skull was exposed, and a 5 μL injection of a 3 pmol solution of GR-73632 (in distilled water; synthesized by Hoffmann-La Roche) was administered via icv injection (directly through the skull at the Bregma reference point) at a depth of 4.5 mm. The dose of GR-73632 used induces stereotypical behaviour patterns, such as foot tapping [59]. Thirty seconds after injection of GR-73632, the needle was removed, local anesthetic was liberally applied to the wound and the incision was closed with wound clips. Following recovery from the anesthesia (recovery of the righting reflex), a 5 min observation period was allocated per study and the amount of time gerbils spent tapping their hind paws was measured.

### 4.5. Induction of Traumatic Brain Injury

Injury was induced in halothane-anesthetized (ProVet) male Sprague Dawley rats (400 ± 20 g) using acceleration-induced impact TBI [29,93]. This model involves impacting a 10 mm diameter, 3 mm thick stainless steel disc fixed centrally to the exposed skull between the lambda and bregma with an accelerating impactor. The impactor was a 450 g brass weight dropped from a height of 2 m. Earlier studies have shown that the impact acceleration injury produces diffuse axonal injury, oedema, BBB opening, Mg^2+^ decline, and moderate-to-severe neurologic deficits [28,29,60]. All animals were fed and watered ad libitum before the induction of injury. During both surgery and the immediate recovery phase, rat rectal temperature was maintained using a thermostatically controlled heating pad. Immediately after injury, animals were manually ventilated until stable respiration was restored, usually in <5 min. After injury, all wounds were sutured. Blood pressure was monitored, both before and after injury, using a femoral arterial line. No significant changes in MABP were noted, including with administration of the drug. Blood pressure monitoring was discontinued after the animal was weaned from the anesthesia and demonstrated recovery, whilst temperature control was ceased when animals were returned to their home cages after recovery.

### 4.6. Drug Treatment

Animals were treated after injury with either the NK1R receptor antagonist, EUC-001 (synthesized by Hoffmann-La Roche, Basel, Switzerland), in isotonic aqueous solution, or with an equal volume of MK801 in aqueous solution, or with an equal volume of saline vehicle (vehicle). The dosage of EUC-001 for functional outcome studies was based on the BBB leakage dose–response curves generated using extravasation of Evans blue (EB) (Sigma-Aldrich, St Louis, MO, USA). A further subgroup of animals that were surgically prepared but not injured (shams) were used as controls, as appropriate. For outcome studies, the administration of EUC-001, MK801, or drug vehicle was performed in a blinded manner.

### 4.7. Dose Response for BBB Permeability

The optimal dose of the NK1 receptor antagonist was determined from the level of BBB permeability using EB dye extravasation as the BBB marker [94,95]. Briefly, EUC-001 was administered intravenously 30 min after TBI at doses ranging between 0.03 and 10 mg/kg in six steps (n = 3 per group). EB dye (Sigma-Aldrich, St Louis, MO, USA; 2 mL/kg of 4%) was administered iv at 4.5 h after TBI and left to equilibrate for 30 min. At 5 h after TBI, animals were transcardially perfused with saline to remove intravascular EB dye and decapitated. This timeframe was based upon previous studies, using this model of TBI, which indicated that BBB permeability was maximal around 5–6 h after injury [96,97]. After decapitation, the brain was removed and the cortex was separated and weighed. Although impact acceleration injury does not normally result in intraparenchymal hemorrhage [29,60], brains were examined routinely during dissection to ensure that no significant intraparenchymal hemorrhage was present which could potentially interfere with EB quantitation. The brain tissue was homogenized and protein precipitated with trichloroacetic acid (Sigma-Aldrich, St Louis, MO, USA). Samples were cooled for 30 min and then centrifuged for 30 min at 1000× *g* (Beckman Coulter, Brea, CA, USA). The supernatants were measured at 610 nm for absorbance of EB using a spectrophotometer (Cole-Parmer, Vernon Hills, IL, USA). Evans blue is expressed as mg/mg of the brain tissue against a standard curve.

### 4.8. Oedema Determination

The amount of oedema development was calculated using the wet weight–dry weight method. Animals were divided into three groups (n = 6 per group) and either left uninjured (sham controls) or treated 30 min after injury with EUC-001, or administered an equal volume of saline vehicle 30 min after injury. They were then re-anesthetized with 4% halothane (Pharmachem, Eagle Farm, Qld, Australia) at 6 h after injury and decapitated. Again, this timepoint was based on the results of previous studies using this model of TBI [96]. The brain was removed rapidly from the skull, the olfactory bulbs and the cerebellum were discarded, and the cortex and subcortex were separated. The cortex and subcortex of each rat were placed separately into pre-weighed and labelled glass vials with quick-fit lids (to prevent evaporation) and weighed immediately for wet water content. The vials (glass lids removed) were then placed in an oven at 100 °C for 72 h. Vials and brain segments were then re-weighed to obtain dry weight content. Edema in each brain sample was calculated using the wet–dry method formula:%Water = (Wet Weight − Dry Weight)/Wet Weight × 100

### 4.9. Functional Outcomes

Motor and cognitive outcomes in animals (n = 6 per group) were assessed using the rotarod and Barnes maze, respectively, as described in detail elsewhere [64]. The rotarod consists of a motorized rotating assembly of 18 rods (1 mm in diameter) upon which the animals were placed. The rotational speed of the device was increased from 0 to 30 r.p.m. and the duration (in seconds) when the animals either completed the task, fell from the rods, or gripped the rods and spun for two consecutive revolutions rather than walking actively was recorded as the rotarod score. Cognitive function was assessed using the Barnes maze as described in the Results section [62,64]. After activating the bright lights and aversive auditory stimulus, the latency (in seconds) for the animal to locate and enter the escape tunnel was recorded. All animals were pretrained on both functional outcome tests twice per day over 5 days before injury to establish a normal, uninjured baseline.

### 4.10. Statistical Analysis

All data are expressed as mean and SEM, with the exception of the functional outcome data, and were analysed for statistical significance using one-way ANOVA (analysis of variance) followed by Student–Neuman–Keuls tests (GraphPad Prism version 10.2.1, La Jol-la, CA, USA). Functional outcome data were analyzed by repeated measures 2-way ANOVA followed by Student–Neuman–Keuls tests.

## 5. Conclusions

EUC-001 represents a selective, high-affinity antagonist of the NK1R. Unlike many other agents in this class, it has a reasonable degree of aqueous solubility whilst still retaining good brain penetration and CNS activity. This solubility makes it suitable for iv administration. Using a rodent model of severe TBI, post-injury administration of EUC-001 was able to help maintain BBB integrity and ameliorate post-injury cerebral oedema. Given that cerebral oedema is considered to play a critical role in mortality during the acute injury phase, this activity of EUC-001 correlates with its significant beneficial effects on the mortality rate. In addition to these acute effects, EUC-001 was also found to confer longer-term functional benefits, both in relation to motor and cognitive function. Given that EUC-001 meets both the pharmacodynamic and pharmacokinetic requirements of a clinical candidate for a novel TBI agent, it has now progressed to clinical trial (eudract_number: 2017-004890-15).

## Figures and Tables

**Figure 1 ijms-25-03535-f001:**
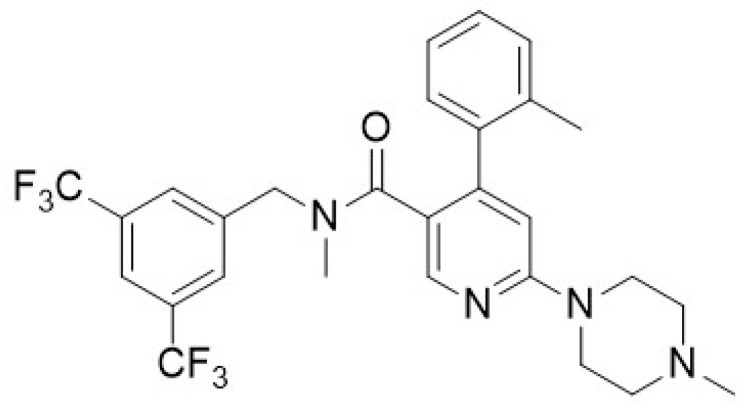
Chemical structure of EUC-001.

**Figure 2 ijms-25-03535-f002:**
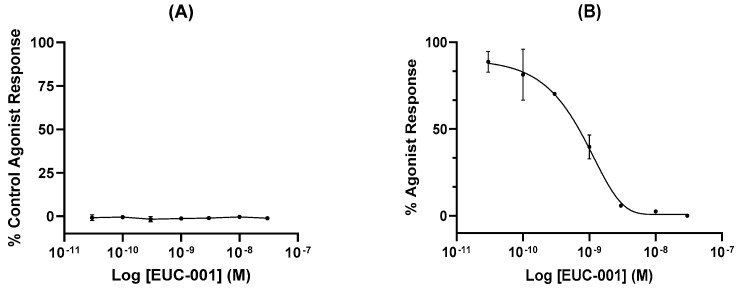
(**A**) The agonist activity of EUC-001 as a percentage of the response induced by 100 nM concentration of the selective NK1R agonist, [Sar^9^, Met(O_2_)^11^]-SP. (**B**) The NK1R antagonist activity of EUC-001, expressed as a percentage change in the response induced by 3 nM [Sar^9^, Met(O_2_)^11^]-SP. Results are expressed as mean ± SEM.

**Figure 3 ijms-25-03535-f003:**
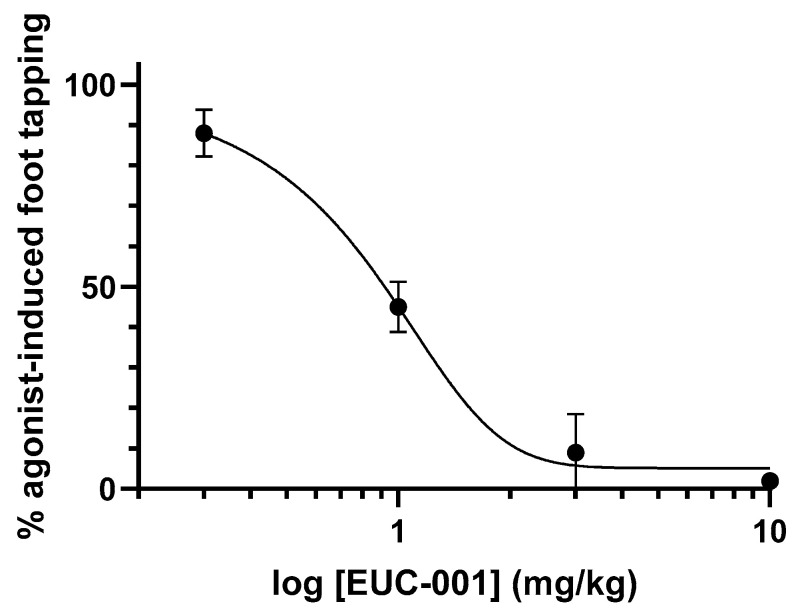
The ability of EUC-001 to inhibit the foot-tapping response induced by the selective NK1R agonist, GR73632. A range of concentrations of EUC-001 was administered *po* 2 h prior to an icv injection of GR73632 (3 pmol/5 µL). Results are expressed as mean ± SEM.

**Figure 4 ijms-25-03535-f004:**
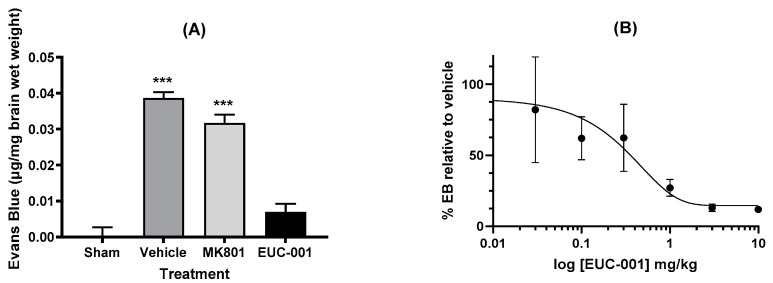
The ability of EUC-001 to reduce BBB permeability, as measured by Evan’s blue leakage, following TBI. (**A**) The effect of administering drug vehicle (saline), MK801 (10 mg/kg), or EUC-001 (10 mg/kg) 30 min post-injury. Evan’s blue was administered 4.5 h after injury, and tissues collected 30 min after Evan’s blue administration. Results are expressed as mean ± SEM; *** *p* < 0.001 relative to Sham and EUC-001 (n = 6). (**B**) Dose-dependent ability of EUC-001, administered 30 min post-injury to inhibit Evan’s blue leakage relative to drug vehicle (saline). Again, Evan’s blue was administered 4.5 h after injury, and tissues were collected 30 min later. Results are expressed as mean ± SEM (n = 3).

**Figure 5 ijms-25-03535-f005:**
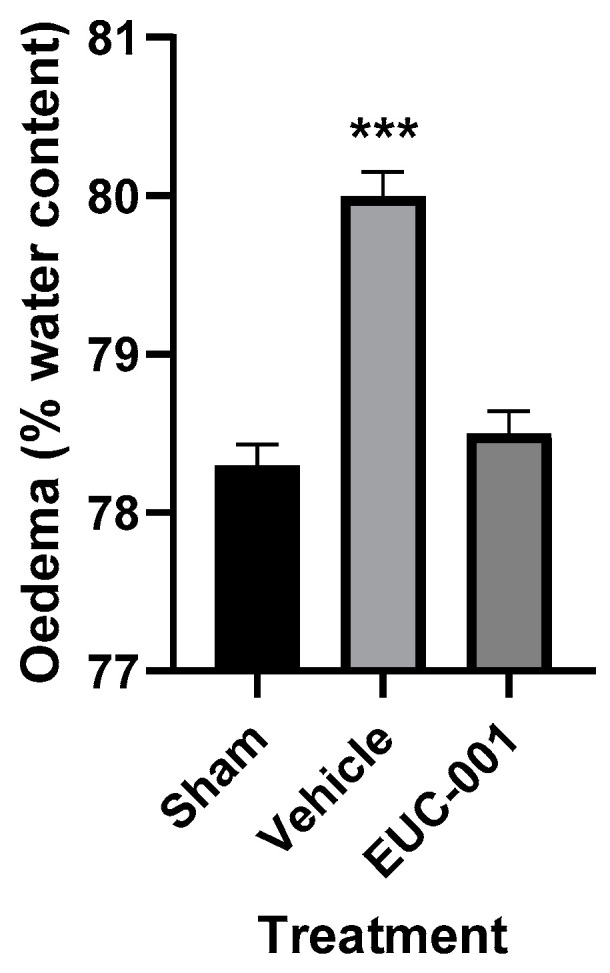
The effect of post-injury (30 min) administration of EUC-001 (10 mg/kg) or drug vehicle (saline) on cerebral oedema, as determined by the wet weight–dry weight method, relative to sham-injured animals. Tissues were collected 6 h after induction of TBI. Results are expressed as mean ± SEM; *** *p* < 0.001 relative to Sham and EUC-001 (n = 6).

**Figure 6 ijms-25-03535-f006:**
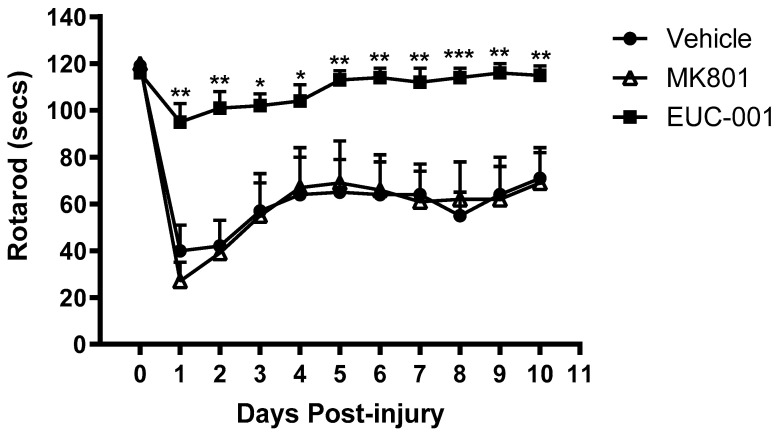
The effect of post-injury drug treatment on motor function as determined by the rotarod test (secs) for 10 days following injury. Either drug vehicle (saline), MK801 (10 mg/kg), or EUC-001 (10 mg/kg) were administered 30 min post-injury. Results are expressed as mean + SEM; * *p* < 0.05, ** *p* < 0.01, and *** *p* < 0.001 relative to drug vehicle and MK8011 (n = 6).

**Figure 7 ijms-25-03535-f007:**
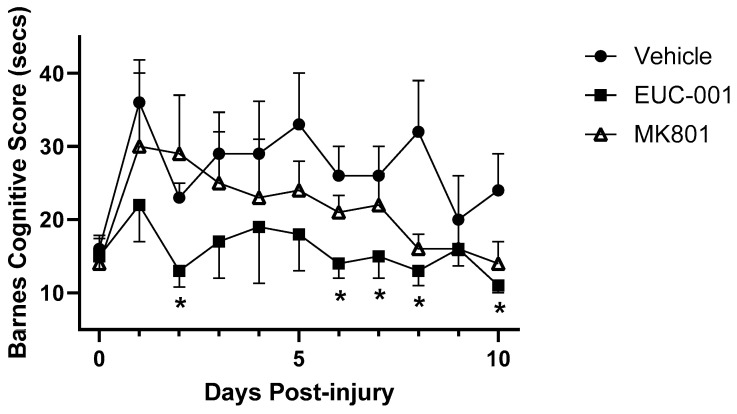
The effect of post-injury drug treatment on cognitive function as determined by the Barnes maze cognitive score (secs) test for 10 days following injury. Either drug vehicle (saline), MK801 (10 mg/kg), or EUC-001 (10 mg/kg) were administered 30 min post-injury. Results are expressed as mean + or − SEM; * *p* < 0.05 relative to drug vehicle (n = 6).

**Table 1 ijms-25-03535-t001:** The ability of EUC-001 to antagonize non-tachykinin cell-surface receptors, expressed as the IC_50_ concentration (M). The receptor subtypes studied were identified as a result of EUC-001 binding activity being detected as part of a broader screening assay.

Receptor	Subtype	IC_50_ (M)
Serotonin	5-HT1B	6.1 × 10^−6^
	5-HT2A	1.5 × 10^−5^
	5-HT5A	4 × 10^−6^
Adenosine	A2A	3 × 10^−5^
	A3	Not calculable
Dopamine	D1	1.6 × 10^−5^
	D2S	Not calculable
Histamine	H2	1.6 × 10^−5^
Melanocortin	MC4	1.3 × 10^−5^
Muscarinic cholinergic	M1	3.3 × 10^−6^
	M2	3 × 10^−5^
	M3	3.3 × 10^−6^

## Data Availability

The majority of the data presented in this study is available upon request from the corresponding author. However, restrictions apply to the availability of some data that were obtained from Roche and are only available from the authors with the permission of Roche.

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
