# Peer review of "Identification of an Intravenous Injectable NK1 Receptor Antagonist for Use in Traumatic Brain Injury"

_ijms, 2024, doi:10.3390/ijms25063535_

Round 1
Reviewer 1 Report
Comments and Suggestions for Authors
Well-structured and designated research.
Including the data on receptor subtype identification would improve manuscript (or may be reference, if it is already published elsewhare).
References should be formatted according to Journal rules.
Author Response
Dear Reviewer,
Firstly we would like to thank you for your time in reviewing our manuscript, and providing us with your constructive feedback.
In terms of providing some background to the receptor subtypes, and the decision to target the NK1 receptor, we have amended that paragraph as follows:
“From knowledge of peripheral inflammatory mechanisms, the pro-inflammatory tachykinin peptide, substance P (SP) was identified as a potential target for a post injury treatment [16, 18]. Clinical data also supports a role for SP in TBI, with strong correlations between SP levels and patient outcomes [22-24]. There are three primary subtypes of the tachykinin receptors, namely NK1, NK2 and NK3 receptors. Whilst there is a potential for cross reactivity with the endogenous ligands and the different receptor subtypes, SP binds preferentially to the NK1 receptor (NK1R), whilst neurokinin A and neurokinin B bind preferentially to the NK2 and NK3 subtypes respectively [25]. As a result, the NK1R was considered an appropriate target to antagonize the actions of SP.”
In terms of the reference style, we will try and identify where the issue might be since the manuscript was prepared using the Journal's template.
Kindest regards,
Alan and Robert
Reviewer 2 Report
Comments and Suggestions for Authors
This well-written study demonstrates that the post-injury administration of EUC-001 improves behavioral deficits, restores BBB function, ameliorates cerebral edema and reduces mortality after the impact acceleration model in rats.
While this study appears well done and clearly detailed, some issues detract from the manuscript’s overall impact.
Specifically:
- Why were animals sacrificed at 6 h after TBI for brain edema measurements? 6h hours are too early to assess edema, as vasogenic and intracellular edema increases by 12 h after the rat model of TBI. At this early time point, if the drug were slowing down the resolution of edema, you would falsely assume it was neuroprotective.
- The same comment is for BBB permeability measurement.
- There is no information that physiological parameters were measured in this study (rectal and head temperatures, blood pressure, etc.).
- Was there any control of rectal temperature and body weight after the recovery period?
- Figure legends should include info on when data was collected.
Author Response
Dear Reviewer,
Firstly, we would like to thank you for the time you took to review our manuscript, and to provide us with your constructive feedback.
In terms of some of the specific points you raised:
- Why were animals sacrificed at 6 h after TBI for brain edema measurements? 6h hours are too early to assess edema, as vasogenic and intracellular edema increases by 12 h after the rat model of TBI. At this early time point, if the drug were slowing down the resolution of edema, you would falsely assume it was neuroprotective.
- The same comment is for BBB permeability measurement.
You are correct in stating that some TBI models result in later oedema. However, our previous studies have shown that oedema and BBB permeability in this model of TBI is maximal at 5-6h (O’Connor, Cernak, Vink 2006, Acta Neurochir 96, 121-4). Habgood et al (2007; Eur J Neurosci, 25, 231-8) also demonstrated that maximal BBB permeability after TBI occurs in the first few hours after TBI (~5h) and reduces thereafter. Hence, our studies have focused on the 5 - 6 hour timepoint. Given your valid comment, we have included a “justification” of our approach for this model of TBI in the manuscript. In addition, we have included the timepoints of data collection in the figure legends as requested.
- There is no information that physiological parameters were measured in this study (rectal and head temperatures, blood pressure, etc.).
- Was there any control of rectal temperature and body weight after the recovery period?
In all studies, rectal temperature was maintained using a thermostatically controlled heating pad. Blood pressure before and after injury was monitored using a femoral arterial line. No significant changes in MABP were noted, including with administration of the drug. Blood pressure monitoring was discontinued after the animal was weaned from the anaesthesia and demonstrated recovery, and temperature control was ceased when animals were returned to their home cages after recovery. We have added this information to the methods section.
We thank you again for your time.
Kindest regards,
Alan and Robert
Reviewer 3 Report
Comments and Suggestions for Authors
The manuscript „Identification of an intravenous injectable NK1 receptor antagonist for use in traumatic brain injury” by Robert Vink and Alan Nimmo is an elegant success story on exploring EUC-001as a selective NK1 antagonist in traumatic brain injuries.
The presentation of physiological consequences of traumatic brain injuries and the search for molecular targets and active substances is written in a very interesting, logical way, however, the rationale of selection of EUC-001 is based on solubility and CNS penetration, as it was a part of bigger screening at Roche. Therefore the scientific curiosity on range of competing compounds is not satisfied. It may be a good idea to include a structure of EUC-001 as a figure, the chemical name is not overly complicated, but a panel showing structures of mentioned NK1 antagonists could be interesting.
The vehicle used in the text (captions, experimental description) should be specified (saline in all cases?).
The effect of MK801, described in lines 411-415, indicates another possible aspect of receptors involved in TBI. This observation is worth discussing in more details.
The sources of chemicals and equipment are mentioned only in some cases, please provide a separate “Materials description” or add missing information in experimental description. This includes information on EUC-001.
Minor issues:
In the chemical names, capital N is usually presented in italics
The multiplication sign in Ki is based on “x” and not the appropriate sign “×”, is it convenience or decision?
There are inconsistencies in presentation of abbreviations and units, some examples: (line 142: NK2 and NK 3), IC50 and IC50 or p.o. (line 184) and po (line 437), 50mM and 0.6 nM, SEM and s.e.m (line 554). The name of EUC-001 is presented in a different way in line 438.
Figure 1: the length of X axis is different, please reconsider.
In general, in figures the SEM values are visible for only some points, and in Fig. 5 only + SEM is offered (for aesthetic purposes?), in Fig. 6, + or – SEM is included.
The caption of figure 1 ends with two dots, information on screening procedure may be added.
The description of Barnes maze is repeated in Results and Experiment description. As results precede experimental part, it helps in understanding the research, therefore some modification to experimental part (reference?) may be considered.
Author Response
Dear Reviewer,
Firstly, we would like to thank you for the time you took to review our manuscript, and to provide us with your detailed and constructive feedback.
In terms of some of the specific points you raised:
- The presentation of physiological consequences of traumatic brain injuries and the search for molecular targets and active substances is written in a very interesting, logical way, however, the rationale of selection of EUC-001 is based on solubility and CNS penetration, as it was a part of bigger screening at Roche. Therefore the scientific curiosity on range of competing compounds is not satisfied. It may be a good idea to include a structure of EUC-001 as a figure, the chemical name is not overly complicated, but a panel showing structures of mentioned NK1 antagonists could be interesting.
This is a valid point, that requires better elaboration in the manuscript. Roche undertook a very large synthesis and screening program that was designed to identify potential lead candidates that could be developed for the indication of depression and anxiety. As such, the program was based around developing an oral agent. The current project involved retrospective data mining within that existing bank of compounds to see if there were any agents that could meet the requirements for TBI, namely an iv injectable agent with good brain penetration. Of that bank of compounds only EUC-001 (and a metabolite of it) fitted those requirements.
We will amend the text to clarify these points and include the structure of EUC-001 as a figure. The publication by Torsten Hoffmann (2006) provides a good description of the Roche screening program.
- The vehicle used in the text (captions, experimental description) should be specified (saline in all cases?).
Yes, the vehicle was normal saline. The text has been amended to indicate this.
- The effect of MK801, described in lines 411-415, indicates another possible aspect of receptors involved in TBI. This observation is worth discussing in more details.
Thank you for this suggestion. We have added a short discussion of the NMDA antagonist issue, including the postulated dual deleterious and beneficial effects.
“Glutamate is a key excitatory transmitter in the CNS. However, events like trauma or cerebral ischaemia can lead to excessive glutamate release, leading to an excitotoxic action, which can be ameliorated by an NMDA antagonist [86]. However, glutamate appears to fulfil a dual role after injury, and may play a role in the CNS recovery process [87]. This has been postulated as one of the reasons that clinical trials of NMDA antagonists in TBI and stroke have failed [87, 88]”
- The sources of chemicals and equipment are mentioned only in some cases, please provide a separate “Materials description” or add missing information in experimental description. This includes information on EUC-001.
We have added the additional information to the experimental description.
Minor issues:
- In the chemical names, capital N is usually presented in italics
Yes, we have corrected this.
- The multiplication sign in Ki is based on “x” and not the appropriate sign “×”, is it convenience or decision?
This has been amended.
- There are inconsistencies in presentation of abbreviations and units, some examples: (line 142: NK2 and NK 3), IC50 and IC50or o. (line 184) and po (line 437), 50mM and 0.6 nM, SEM and s.e.m (line 554). The name of EUC-001 is presented in a different way in line 438.
We have corrected these inconsistencies.
- Figure 1: the length of X axis is different, please reconsider.
We have tried to balance these. (Now Fig 2)
- In general, in figures the SEM values are visible for only some points, and in Fig. 5 only + SEM is offered (for aesthetic purposes?), in Fig. 6, + or – SEM is included.
Where possible, we have included +/- SEM. For some points, these are not visible due to the size of the SEM and the scale of the graph. For figures 5 and 6, a conscious decision was made to use only + or – for the purpose of both aesthetics and clarity.
- The caption of figure 1 ends with two dots, information on screening procedure may be added.
We have corrected this. In terms of the screening procedure we chose not to include this because the screen was designed to deliver orally-active NK1 antagonists. What we undertook was a retrospective search for a TBI candidate.
- The description of Barnes maze is repeated in Results and Experiment description. As results precede experimental part, it helps in understanding the research, therefore some modification to experimental part (reference?) may be considered.
We agree that it makes sense not to duplicate this, so we have amended the experimental section.
We thank you again for your time.
Kindest regards,
Alan and Robert
Reviewer 4 Report
Comments and Suggestions for Authors
ijms-2907952, Identification of an intravenous injectable NK1 receptor antagonist for use in traumatic brain injury
The manuscript effectively communicates the key findings of the study and the potential of EUC-001 as a therapeutic intervention for TBI. The research is solid and presented in a clear manner. There are some small issues that could be improved.
The introduction section should present the chemical structure for the mentioned NK1 antagonists and also for EUC001. The authors should add details on the drug design process that prompted the development of this lead Add some reference for the information on rows 122-128.
The authors should eliminate the company names. They are not relevant for the work presented here.
The manuscript should be presented in the journal’s style
The manuscript should be unitary in style. See s.e.m. and SEM
Author Response
Dear Reviewer,
Firstly, we would like to thank you for the time you took to review our manuscript, and to provide us with your detailed and constructive feedback.
In terms of some of the specific points you raised:
- The introduction section should present the chemical structure for the mentioned NK1 antagonists and also for EUC001. The authors should add details on the drug design process that prompted the development of this lead Add some reference for the information on rows 122-128.
We agree with your comment. Because the actual screen was designed to identify orally-active NK1 antagonists for the depression market, rather than describe the screen itself, we have elaborated on the retrospective data interrogation that was performed. This is now described between the introduction and the results.
"Given the range of analogues developed by this screening program it was decided to perform a retrospective data mine into this bank of compounds to see whether any of the agents possessed the pharmacodynamic and physicochemical properties that would make them suitable for development as a clinical agent for TBI. Interrogation of the available data revealed that one compound (RO-0671721; EUC-001) possessed the required characteristics. The present study describes the pre-clinical characterization of EUC-001 as a potential candidate for development as a treatment for TBI.
- Results
Data interrogation of the bank of NK1R antagonists developed as part of the Roche screening program [56], indicated that one compound, N-(3,5-Bis-trifluoromethyl-benzyl)-N-methyl-6-(4-methyl-piperazine-1-yl)-4-o-tolyl-nicotinamide (EUC-001) possessed the appropriate physicochemical characteristics required (Figure 1). EUC-001 was found to have an aqueous solubility of 0.5mg/ml as a free base, whilst the hydrochloride salt could be dissolved at a concentration of 1.5mg/mL, indicating the potential to administer it in an iv formulation. Whilst EUC-001 has greater aqueous solubility than many NK1 antagonists, it was found to still possess good CNS penetration, with a brain/plasma ratio of 3/7."
We have now included the structure of EUC-001 as Figure 1
- The authors should eliminate the company names. They are not relevant for the work presented here.
We have tried to remove as much of the company references as possible, whilst still retaining the “sense” of the project history.
- The manuscript should be presented in the journal’s style
We have tried to conform to the style through the use of the journal’s electronic template
- The manuscript should be unitary in style. See s.e.m. and SEM
We have tried to remove all inconsistencies with abbreviations.
We thank you again for your time.
Kindest regards,
Alan and Robert